# Understanding and Scaling Collaborative Filtering Optimization from the Perspective of Matrix Rank

## Abstract

Collaborative Filtering (CF) methods dominate real-world recommender systems given their ability to learn high-quality, sparse ID-embedding tables that effectively capture user preferences. These tables scale linearly with the number of users and items, and are trained to ensure high similarity between embeddings of interacted user-item pairs, while maintaining low similarity for non-interacted pairs. Despite their high performance, encouraging dispersion for non-interacted pairs necessitates expensive regularization (e.g., negative sampling), hurting runtime and scalability. Existing research tends to address these challenges by simplifying the learning process, either by reducing model complexity or sampling data, trading performance for runtime. In this work, we move beyond model-level modifications and study the properties of the embedding tables under different learning strategies. Through theoretical analysis, we find that the singular values of the embedding tables are intrinsically linked to different CF loss functions. These findings are empirically validated on real-world datasets, demonstrating the practical benefits of higher stable rank – a continuous version of matrix rank which encodes the distribution of singular values. Based on these insights, we propose an efficient warm-start strategy that regularizes the stable rank of the user and item embeddings. We show that stable rank regularization during early training phases can promote higher-quality embeddings, resulting in training speed improvements of up to **65.9%**. Additionally, stable rank regularization can act as a proxy for negative sampling, allowing for performance gains of up to **21.2%** over loss functions with small negative sampling ratios. Overall, our analysis unifies current CF methods under a new perspective – their optimization of stable rank – motivating a flexible regularization method that is easy to implement, yet effective at enhancing CF systems. Code provided at *Repo Link*.

## CCS Concepts

• **Information systems → Retrieval models and ranking**.

## Keywords

Collaborative Filtering, Recommendation, Matrix Rank, Scalability

**ACM Reference Format:**
Anonymous Author(s). 2018. Understanding and Scaling Collaborative Filtering Optimization from the Perspective of Matrix Rank. In *Proceedings of (WWW '25)*. ACM, New York, NY, USA, 14 pages. https://doi.org/XXXXXXX.XXXXXXX

## 1 Introduction

Across the web, recommender systems play a pivotal role in delivering personalized user experiences [9, 13, 26, 32, 40]. From e-commerce platforms offering tailored product suggestions to music streaming services organizing personalized playlists, these systems have become integral to navigating the vast amount of online information [10, 19, 20, 24]. At the forefront of recommender systems is collaborative filtering (CF), a technique that predicts unknown user preferences from the known preferences of a set of users [3, 30, 33]. One of the most prominent variants of CF is matrix factorization (MF) which learns embeddings for each user and item from historical user-item interaction data [15, 16, 23, 35]. Once trained, MF-based models are able to efficiently filter and rank content for each user, offering a curated and personalized experience. [6, 44].

With advancements in CF, such as deep neural networks [4, 36], message passing [15, 22, 37], and loss function design [25, 35, 39], the computational demands to train CF models has risen considerably [15, 35, 37]. These demands are further exacerbated as the number of users and items increases, leading to a considerable rise in training time [34]. The challenge of scaling to large user and item sets is most apparent in modern loss functions, where recent losses, such as DirectAU [35] and MAWU [25], scale quadratically with the number of users and items. However, it is also well known that computationally heavy losses, such as DirectAU, MAWU, and Sampled Softmax (SSM) [39] significantly out-perform lighter losses, such as Bayesian Personalized Ranking (BPR) [30]. To manage the computational cost, compromises are often made on either the system's parameterization or architecture, however this also risks reducing personalization and overall performance [7, 12, 17]. Thus, maintaining high performance while reducing computational burden presents a significant challenge in modern CF systems.

Focusing on loss function design, previous work has highlighted that many CF losses, including BPR, SSM, and DirectAU, differ primarily by their regularization strength [25]. Moreover, this regularization is largely related to the number of negative samples considered during training. Thus, it would appear that the trade-off between performance and run-time is fundamental, given more negative samples incurs a higher computational overhead. However, we question if it is possible to attain a cheaper proxy for negative sampling that can be leveraged as an alternative regularization during training. To answer this question, we focus on the one shared aspect amongst all of the aforementioned designs: *the user and item embedding matrices*. This then leads us to consider a more fundamental question:

***Are there intrinsic properties of the embedding matrices that contribute to high-performing CF systems?***

By identifying such properties, we are able to examine why certain learned matrices perform better than others, and elucidate candidate matrix properties that can be leveraged as priors on the training process to improve embedding quality.

Through our analysis, we uncover that the stable rank[1] [18, 31] (a continuous variant of rank) of the user and item embedding matrices tends to positively correlate with the negative sampling rate, as seen in the difference between systems trained with BPR and DirectAU. We empirically demonstrate this result by examining the optimization trajectory of stable rank across various datasets and loss functions, also drawing a link between stable rank and performance. We then provide a theoretical explanation for how varying levels of stable rank emerge from different loss functions, linking CF optimization to the singular values of the user and item matrices. Based on these findings, we propose a stable rank regularization which is utilized as a warm-start mechanism for CF training, acting as a cost-effective proxy for negative sampling.

Focusing on BPR, SSM, and DirectAU as a family of losses which induce different levels of negative sampling-based regularization, we study how stable rank regularization can: (a) replace expensive full negative sampling, such as in the regularization term of DirectAU, as well as (b) provide the full negative sampling training signal for lighter losses, e.g. BPR. Through empirical analysis, we demonstrate that warm-starting systems trained with DirectAU can save multiple hours of training given the linear scaling with the number of users during warm-start epochs. Additionally, for systems trained with BPR, stable rank regularization achieves significant performance increases with a small increase in computational overhead, given the warm-start epochs approximates more expensive negative sampling strategies. Overall, our analysis unifies common CF training paradigms from the novel perspective of stable rank optimization of the user and item embedding matrices. Given our proposed method is model-agnostic and lightweight, it can be easily applied with minimal overhead, helping to promote scalability in real-world systems. Our contributions are outlined below:

- **Linking Negative Sampling, Matrix Rank, and CF Performance:** We offer the first analysis which formally connects negative sampling with matrix rank. We also demonstrate a correlation between matrix rank and higher performance.
- **Theoretical Analysis on Matrix Rank:** We theoretically relate common CF training paradigms to matrix rank, demonstrating that alignment induces rank collapse in user and item matrices, whereas uniformity promotes rank increase in low rank settings.
- **Warm Start Strategy for Scalable Recommenders:** Using our newfound understanding, we propose a warm-start strategy which acts as a cost-effective proxy for negative sampling, enabling faster learning of high quality embeddings.
- **Extensive Empirical Analysis:** We show that stable rank regularization is able to provide (i) significant speed benefits to more expensive loss functions, e.g. DirectAU, attaining up to a **65.9%** decrease in training time, and (ii) significant performance benefits to light-weight loss functions, e.g. BPR, attaining up to a **21.7%** performance increase.

## 2 Preliminaries and Related Work

### 2.1 Collaborative Filtering

Given an interaction set $\mathcal{E}$ between a set of users $U$ and items $I$, collaborative filtering (CF) learns unique embeddings for each

---

[1]Defined later in Equation (5), Section 2.3.

user $u \in U$ and item $i \in I$ such that the interactions between user and items can be recovered [1]. The matrix which holds the interaction set is denoted $\mathbf{E} \in \mathbb{Z}^{|U| \times |I|}$. The embeddings for the set of users and items are represented through the user and items matrices, $\mathbf{U} \in \mathbb{R}^{|U| \times d}$ and $\mathbf{I} \in \mathbb{R}^{|I| \times d}$, where $d$ is the embedding dimensionality. The most common learning paradigm for CF is matrix factorization (MF), where $\mathbf{U}$ and $\mathbf{I}$ are learned such that $\mathbf{E} \approx \mathbf{U}\mathbf{I}^\top$. Letting $\mathbf{u}$ and $\mathbf{i}$ be the user and item embeddings associated with a user $u$ and item $i$, respectively, the interaction signal under MF is recovered via the dot product between $\mathbf{u}$ and $\mathbf{i}$, i.e. $\mathbf{E}_{u,i} \approx \mathbf{u} \cdot \mathbf{i}^\top$.

Despite MF's effectiveness, the dot product as an interaction function limits expressivity [16]. Thus, variants of MF have proposed using neural networks to introduce non-linearities into the interaction calculation. For instance, one may transform the user and item matrices using a deep neural network (DNN), e.g. with DNNs $F$ and $G$, $\mathbf{E} \approx F(\mathbf{U})G(\mathbf{I})^\top$, or one can parameterize the interaction calculation, letting $\mathbf{E}_{u,i} \approx H(\mathbf{u}, \mathbf{i})$ for DNN $H$ [4, 36, 43]. Recent advancements in graph machine learning have also motivated the development of graph-based CF methods that leverage message passing over the embeddings before the interaction function [11, 15]. While both methods tend to improve the performance of recommender systems over the traditional MF baseline, each introduces a significant computational cost.

### 2.2 Optimization for Collaborative Filtering

Given we cannot directly factor $\mathbf{E}$, an approximate solution can be obtained by formulating a linear least squares objective with respect to the predicted matrix $\hat{\mathbf{E}} = \mathbf{U}\mathbf{I}^\top$. This is formalized as $L = ||\mathbf{E} - \hat{\mathbf{E}}||_F$, where $L$ can be minimized by letting $\hat{\mathbf{E}}$ be the Singular Value Decomposition (SVD) of $\mathbf{E}$. In practice, due to matrix size and overfitting concerns, $L$ is solved through gradient descent. However, the properties of $\mathbf{U}$ and $\mathbf{I}$ learned through gradient descent have yet to be studied, making it unclear how different loss functions benefit CF. The loss functions consdered in this work are outlined with discussion on their trade-offs.

*2.2.1 Bayesian Personalized Rank (BPR).* One of the traditional losses to train MF models is BPR [8, 30]. Rather than predicting the exact interaction value, BPR optimizes for ranking by maximizing the margin between preferred and non-preferred items for each user. This is achieved through a pair=wise loss which maximizes the distance between interacted and non-interacted samples. Formally, for a set of user-item triplets, where each triplet is comprised of a user $u$, interacted item $i$, and non-interacted item $i'$, the loss is:

$$L_{BPR} = \sum_{(u,i,i') \in \mathcal{D}} \ln \sigma(\hat{e}_{u,i} - \hat{e}_{u,i'}), \tag{1}$$

where $\hat{e}_{u,i}$ is the similarity between $u$ and $i$. In a classic MF setting, the similarity is just the dot product between $\mathbf{u}$ and $\mathbf{i}^\top$. By focusing on the relative order of items, rather than their absolute scores, BPR enhances the ranking quality of recommenders.

*2.2.2 Sampled Softmax (SSM).* Set-wise losses generalize pair-wise losses, like BPR, by considering $k$ negative samples for each user-item pair. A common variant of set-wise loss in recommendations is SSM [28, 39]. SSM reduces the large set of negative samples (i.e., the rest of the items not in the positive set for a user) to a subset of negative samples which need to be ranked lower than the positive

items. When the number of negative samples is one, this reduces to BPR. SSM is expressed as:

$$L_{SSM} = -\log \frac{\exp(\hat{e}_{u,i})}{\exp(\hat{e}_{u,i}) + \sum_{i' \in S} \exp(\hat{e}_{u,i'})}, \tag{2}$$

where $S$ is the set of negative samples. By using a subset of the possible negative samples, SSM is able to retain competitive efficiency while generally achieving higher performance than BPR [29].

*2.2.3 DirectAU.* DirectAU is a loss function for CF that directly optimizes both similarity between interacted users and items (alignment) and dispersion between user-user or item-item pairs (uniformity) [35]. DirectAU leverages principles from contrastive losses which remove the need for explicit negative sampling, as in BPR and SSM. The alignment component of DirectAU is specified as:

$$L_{align} = \sum_{(u,i) \in \mathcal{E}} \|\mathbf{u} - \mathbf{i}\|^2, \tag{3}$$

where $(u, i)$ are observed user-item interactions. Alignment aims to promote similarity between embeddings for users and items which share an interaction. To ensure embeddings do not overfit to the historic user-item pairs, the uniformity component of DirectAU promotes that user and item representations be dispersed on the $d$-dimensional hyperspheres. This term is formulated as:

$$L_{uniform} = \log \sum_{(u,u') \in U} e^{-2\|\mathbf{u} - \mathbf{u}'\|^2} + \log \sum_{(i,i') \in I} e^{-2\|\mathbf{i} - \mathbf{i}'\|^2}. \tag{4}$$

Note that the embeddings are normalized for the alignment and uniformity term. The DirectAU loss weights these two terms with trade-off parameter $\gamma$. DirectAU offers several advantages by helping to prevent embedding collapse; however, the uniformity term introduces significant computational overhead, scaling as $O(|U|^2 d)$. While some works have offered improvements to DirectAU, this has centered around mitigating popularity bias and generally requires even more parameters than the original formulation [29].

*2.2.4 Relationship Between the Losses.* It is well established that DirectAU tends to outperform SSM, and SSM tends to outperform BPR [35]. More recent work has highlighted that uniformity in DirectAU has a relationship with BPR and SSM, where uniformity can be interpreted as considering all possible negative samples [29]. From this perspective, BPR, SSM, and DirectAU can be seen as a family of techniques which consider stronger level of regularization, induced by the number of negative samples. However, the significant increase in complexity introduced by both SSM, when $k$ is large, and DirectAU may be impractical for real-world applications. To mitigate this trade-off, we consider if there is an alternative method to attain regularization conferred by negative sampling that scales more favorably with the size of the user and item sets.

## 2.3 Matrix Rank

The rank of a matrix is used to characterize the dimension of the vector space spanned by the matrix. A generic method to define the rank of a matrix $\mathbf{A}$ is given by $\text{rank}(\mathbf{A}) = |\{\sigma_r | \sigma_r > 0\}|$, where $\sigma_r$ is the $r$-th singular value of $\mathbf{A}$. The singular values can be extracted through the SVD of $\mathbf{A}$, where $\mathbf{A} = \Psi \Sigma \Omega^\top$ and $\Psi$ is the matrix of left singular vectors, $\Sigma$ is the matrix of singular values along the diagonal, and $\Omega$ is the matrix of right singular vectors. In practice, it is common to compute the rank for singular values greater than $\epsilon$, rather than 0, due to numerical precision.

To provide a more comprehensive understanding of the singular values of a matrix, while also alleviating the challenge of setting an appropriate $\epsilon$, the stable rank of a matrix is often utilized in matrix analysis [18, 31]. The stable rank of a matrix $\mathbf{A}$ is defined as:

$$\text{srank}(\mathbf{A}) = \frac{\|\mathbf{A}\|_F^2}{\|\mathbf{A}\|_2^2} = \frac{\sum_{r \in \{1,\dots,\text{rank}(\mathbf{A})\}} \sigma_r^2}{\sigma_1^2}, \tag{5}$$

where $\sigma_r$ is a non-zero singular value of $\mathbf{A}$, sorted in descending order or magnitude, and $\sigma_1$ is the largest singular value of $\mathbf{A}$. Intuitively, $\text{srank}(\mathbf{A})$ can be interpreted as a continuous variant of traditional matrix rank where the relative contribution of a singular value is directly encoded, rather than discretized to 0 or 1. From the perspective of optimization during MF training, stable rank can be used to characterize how effectively the model is utilizing the $d$ dimensions of the embeddings.

## 3 Motivation - Matrix Properties of High-Quality User and Item Matrices

In this section, we study the properties of the user and item embedding matrices learned across different CF methods. Through our findings, we can decipher what constitutes higher-quality embeddings, as measured by performance, and leverage such knowledge during training. To study the properties of the embedding matrices, we begin by simply training an MF model with both BPR and DirectAU across four different benchmarks. We then study the training trajectories for the different models, focusing on how the stable rank changes. The final stable rank of the learned embedding matrices is compared to the performance of the respective model. Our results demonstrate that stable rank tends to be highly correlated with stronger performance between BPR and DirectAU, prompting a deeper study on how stable rank is optimized in the different methods and how it can be utilized to improve training. Exact details for the empirical setup can be found in Appendix C.

## 3.1 Stable Rank Trajectories

In Figure 1, we plot the stable rank trajectories for different benchmark datasets and the BPR and DirectAU loss functions. Initially, both models start at a high stable rank due to random initialization. However, both display significantly different trajectories. BPR tends to decrease in stable rank across epochs, arriving at an overall low stable rank at early stopping. DirectAU is significantly different, where initially the matrices collapse to a lower stable rank, and then later increase after a period of training. As DirectAU utilizes both an alignment and uniformity loss, DirectAU is able to both collapse and disperse the representations of the user and item matrices, as opposed to BPR which tends to prioritize collapse. We hypothesize that these loss behaviors are the direct cause of the different stable rank values, which we validate in subsequent sections.

## 3.2 Relationship between Performance and Stable Rank

With establishing the different stable rank trajectories between BPR and DirectAU, we now consider the quality of the resulting

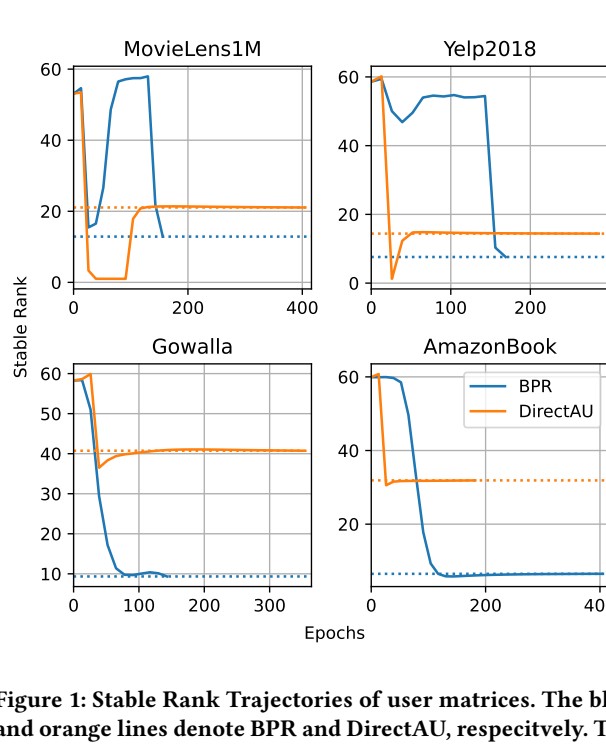

**Figure 1: Stable Rank Trajectories of user matrices. The blue and orange lines denote BPR and DirectAU, respecitvely. The dotted lines denote the final stable rank of the user matrices after training. DirectAU produces higher stable rank. Similar trends are found for the item matrices, as seen in Appendix D.**

**Table 1: NDCG@20 and stable rank for different datasets and losses. Both metrics are reported as an average over three random seeds with standard deviations.**

| Dataset | Loss Function | NDCG@20 | Stable Rank |
|---|---|---|---|
| MovieLens1M | BPR | 0.194 ± 0.0 | 12.921 ± 0.046 |
| | DirectAU | 0.236 ± 0.0 | 21.082 ± 0.015 |
| Yelp2018 | BPR | 0.047 ± 0.001 | 7.471 ± 0.13 |
| | DirectAU | 0.071 ± 0.0 | 14.423 ± 0.019 |
| Gowalla | BPR | 0.092 ± 0.001 | 9.386 ± 0.043 |
| | DirectAU | 0.132 ± 0.001 | 40.681 ± 0.061 |
| AmazonBook | BPR | 0.046 ± 0.0 | 6.5 ± 0.01 |
| | DirectAU | 0.078 ± 0.0 | 31.852 ± 0.033 |

learned user and item embeddings for the different settings. For each dataset and loss, we compare their performance, computed as NDCG@20, and stable rank. In Table 1, we see that in each instance, the higher performing loss function additionally has a higher stable rank, drawing a correlation between the two.

To provide intuition on this behavior, we consider the interaction matrix $\mathbf{E}$ for sets of users $U$ and items $I$. We know that the rank($\mathbf{E}$) $\leq \min(|U|, |I|)$, where the less than condition of the inequality accounts for users with identical item interactions, or items with identical user interactions. As it is common to set the embedding dimension of $\mathbf{U}$ and $\mathbf{I}$, $d$, to be significantly less than $\min(|U|, |I|)$, and given interactions matrices tend to be extremely

sparse minimizing duplicate interaction patterns, we assume the goal is to learn a rank-$d$ approximation of $\mathbf{E}$. The optimal solution then comes from the truncated SVD of $\mathbf{E}$ with $d$ retained singular values. Thus,

$$\mathbf{E}_d = \mathbf{\Psi}_d \mathbf{\Sigma}_d \mathbf{\Omega}_d^\top = (\mathbf{\Psi}_d \mathbf{\Sigma}_d^{\frac{1}{2}})(\mathbf{\Sigma}_d^{\frac{1}{2}} \mathbf{\Omega}_d \top) = \mathbf{U}_d \mathbf{V}_d^\top. \quad (6)$$

Using the Eckart–Young–Mirsky theorem, the error in the rank-$d$ approximation of $\mathbf{E}$ is then given by:

$$\|\mathbf{E} - \mathbf{E}_d\|_F^2 = \|\mathbf{E} - \mathbf{U}_d \mathbf{V}_d^\top\|_F^2 = \sum_{r=d+1}^{\text{rank}(\mathbf{E})} \sigma_r^2. \quad (7)$$

In practice, the matrix $\mathbf{E}$ is often too large to directly compute SVD, thus the true rank($\mathbf{E}$) is unknown. Moreover, even computing a truncated SVD is only possible on small datasets. Thus, $\mathbf{U}_d$ and $\mathbf{V}_d$ are generally approximated via gradient descent. While truncated SVD ensures $\mathbf{U}$ and $\mathbf{I}$ are rank $d$, the chosen gradient descent loss function can incorporate inductive biases which further reduce this rank below $d$, potentially hurting performance given the relation in Equation (7). To thoroughly understand how certain rank values arise, in the next section we offer a rigorous analysis on how different losses impact the gradient descent process and induce different rank properties into the the user and item matrices.

## 4 Theoretical Analysis - Matrix Properties Induced by Different Losses

Despite BPR, SSM, and DirectAU sharing a common underlying optimization paradigm, related to the number of negative samples considered in the training process [29], the implications on the user and item matrices (beyond negative sampling improving performance) remains unclear. Due to this missing connection, it is unclear what properties negative sampling induces into the embedding matrices, and *how* one might create a proxy for negative sampling via priors on the training process. To address this gap, we build upon our established empirical findings on stable rank and carefully study how optimization through different CF losses changes the user and item embedding matrix properties. Specifically, we study the cases of pure alignment optimization (zero negative samples), and pure uniformity optimization (all negative samples), demonstrating that the adjustment of stable rank is intrinsically encoded in the matrix updates. With our newfound theoretical relationship, we propose a training strategy able to induce the benefits of full negative sampling with a significantly smaller computational cost.

### 4.1 Singular Values under Alignment and Uniformity Optimization

Below we offer two theoretical analyses where we study the properties of the user matrix $\mathbf{U}$ after training solely with alignment and solely with uniformity, respectively. For each analysis, we provide the assumptions, our theoretical analysis, and the implications.

*4.1.1 Optimizing Alignment.* We assume user and item embedding matrices $\mathbf{U} \in \mathbb{R}^{n \times d}$ and $\mathbf{I} \in \mathbb{R}^{m \times d}$, as well as an interaction set $\mathcal{E}$, where $(j, l) \in \mathcal{E}$ denotes a user $u_j$ interacted with item $i_l$. For brevity, we focus on a mini-batch of interactions between a set of $r$ users, $\{u_1, ..., u_r\}$, and a particular item $i$. We then compute the

ratio of the first and second singular values for a gradient descent step $t$, denoting the learning rate as $\eta$.

THEOREM 4.1. *Given the initial user embedding matrix, $\mathbf{U}^{(0)}$, with singular values $\sigma_1^{(0)}$ and $\sigma_2^{(0)}$, and the user embedding matrix after $t$ iterations of gradient descent with $L_{align}$, $\mathbf{U}^{(t)}$, with singular values $\sigma_1^{(t)}$ and $\sigma_2^{(t)}$, $\Delta_{ali}^{(t)} = \frac{\sigma_1^{(0)}}{\sigma_2^{(0)}} / \frac{\sigma_1^{(t)}}{\sigma_2^{(t)}}$ is given by:*

$$\Delta_{ali}^{(t)} = \frac{\sigma_1^{(0)}(1-2\eta)^t}{(1-(1-2\eta)^t)\sqrt{r}\|\mathbf{i}\|_2 + \sigma_1^{(0)}(1-2\eta)^t}. \tag{8}$$

The full proof is given in Appendix A.1. When $\eta < \frac{1}{2}$, the first term in the denominator is strictly positive and $\Delta_{ali}^{(t)} < 1$, indicating that the first and second singular values diverge as $t$ increases. Thus, training purely with alignment, i.e. without negative sampling, induces lower stable rank in the user matrix. Similar logic can be applied to the item embedding matrix by flipping the initial notation, indicating a similar decline in stable rank.

*4.1.2 Optimizing Uniformity.* As uniformity operates solely on either the user or item embedding matrix, WLOG we focus only on the user embedding matrix $\mathbf{U} \in \mathbb{R}^{n \times d}$. Similar to alignment, we compute the ratio of the first and second singular values for a gradient descent step $t$.

THEOREM 4.2. *Given the user embedding matrix optimized via $L_{uniform}$ at gradient step $t$ and $t+1$, with singular values $\sigma_1^{(t)}$, $\sigma_2^{(t)}$ and $\sigma_1^{(t+1)}$, $\sigma_2^{(t+1)}$, respectively, $\Delta_{uni}^{(t)} = \frac{\sigma_1^{(t)}}{\sigma_2^{(t)}} / \frac{\sigma_1^{(t+1)}}{\sigma_2^{(t+1)}}$ is given by:*

$$\Delta_{uni}^{(t)} = \frac{\sigma_1^{(t)}(\alpha\sigma_2(\delta\mathbf{U}^{(t)}) + \sigma_1^{(t)})}{\sigma_2^{(t)}(\alpha\sigma_1(\delta\mathbf{U}^{(t)}) + \sigma_1^{(t)})}, \tag{9}$$

where $\alpha = \eta 4e^{-4}\sqrt{nd}$, $\delta\mathbf{U}^{(t)} \in \mathbb{R}^{n \times n}$ is the matrix of L1 distances between pairs of rows in $\mathbf{U}^{(t)}$, and $\sigma_j(\delta\mathbf{U}^{(t)})$ is the $j$-th singular value of $\delta\mathbf{U}^{(t)}$. The full proof is given in Appendix A.2. $\Delta_{uni}^{(t)} > 1$ when $\frac{\sigma_1^{(t)}}{\sigma_2^{(t)}} > \frac{\sigma_1(\delta\mathbf{U}^{(t)})}{\sigma_2(\delta\mathbf{U}^{(t)})}$. If we consider a rank-2 user embedding matrix, where the individual user vectors deviate by angle $\epsilon$, $\frac{\sigma_1^{(t)}}{\sigma_2^{(t)}} \approx \frac{1}{\epsilon}$ and $\frac{\sigma_1(\delta\mathbf{U}^{(t)})}{\sigma_2(\delta\mathbf{U}^{(t)})} \approx \frac{r-1}{\sqrt{r}}$. Thus, as $\epsilon$ tends towards 0, $\Delta_{uni}^{(t)}$ is greater than 1, indicating that uniformity promotes higher stable rank as the embeddings with the matrix become more aligned.

# 5 Expediting Collaborative Filtering Training with Stable Rank Regularization

Given our analysis thus far, we have evidence that (a) the stable rank of the user and item matrices influences model performance, and (b) negative sampling-based regularization strategies intrinsically induce higher stable rank. Thus, our goal is to directly induce the stable rank property within the training process, rather than indirectly optimize it through a more costly regularization term, like uniformity. We begin by introducing our new loss term, stable rank regularization, and then discuss how we use it during training.

## 5.1 Stable Rank Regularization

We formulate the stable rank regularization as the stable rank calculation scaled relative to the max possible rank of the matrix. The scaling allows for the stable rank loss to be between 0 and 1, making it easier to balance with other loss terms. Specifically, given $\mathbf{A} \in \mathbb{R}^{n \times m}$, the regularization is formulated as:

$$L_{\text{srank}} = \frac{\|\mathbf{A}\|_F^2}{\|\mathbf{A}\|_2^2 \max(n, m)}. \tag{10}$$

Notably, $L_{\text{srank}}$ is model-agnostic, and can be amenable to any ID embedding training strategy by co-optimizing it with some similarity-inducing loss, e.g. $L_{\text{align}}$. We optimize with respect to $-\gamma_{sr} L_{\text{srank}}$ to retain higher stable rank, where $\gamma_{sr}$ is the weighting parameter between the chosen similarity loss and $L_{\text{srank}}$. We can additionally express $\nabla_{\mathbf{A}} L_{\text{srank}}(\mathbf{A})$ as:

$$\nabla_{\mathbf{A}} L_{\text{srank}}(\mathbf{A}) = \frac{2(\mathbf{A} - \text{srank}(\mathbf{A})\sigma_1\psi_1\omega_1^\top)}{\|\mathbf{A}\|_2^2}, \tag{11}$$

where $\sigma_1$ is the largest singular value of $\mathbf{A}$, $\psi_1$ is the left singular vector corresponding to $\sigma_1$, and $\omega_1$ is the right singular vector corresponding to $\sigma_1$. Similar to uniformity, we row-normalize $\mathbf{A}$.

*5.1.1 Relation to Other Methods.* The proposed stable rank regularization has relation to some methods found in self-supervised learning (SSL). For instance, the Barlow Twins loss aims to maximize the off-diagonals of the the correlation matrix between perturbed samples, mitigating dimensionality collapse [42]. However, this is highly coupled to the SSL setting and not directly amenable to CF. More general contrastive losses have also discussed the benefit of optimizing spectral properties of embeddings [21]. In the context of CF, the newly proposed nCL loss has begun to explore these principles, optimizing for compact and high-dimensional clusters for users and items [2]. Yet, the loss is only motivated empirically by performance and cannot be applied to pre-existing systems.

*5.1.2 Computational and Memory Complexity:* Computing $\|\mathbf{A}\|_F^2$ requires iterating over all elements within $\mathbf{A}$, thus scaling as $O(nm)$. $\|\mathbf{A}\|_2^2$ requires solving for the largest eigenvalue of $(\mathbf{A}^\top\mathbf{A})^{\frac{1}{2}}$, which scales as $O(nm^2)$ given the number of embedding dimensions is small ($n \gg m$). In comparison, uniformity scales as $O(n^2m)$ given the need to compute all pairwise similarities between users or items. As the intermediary matrices must be retained for gradient computation, as seen in Equations (12) and (13), the memory requirements follow similar scaling properties as the computation requirements. Thus, stable rank regularization is not only faster, but allows for training with larger datasets given a fixed memory size.

## 5.2 Training with Stable Rank Regularization

Despite establishing that stable rank is optimized within CF training, it is still unclear to what extent stable rank can be used as a direct proxy for negative sampling during training. Thus, we first study the relationship between stable rank and uniformity optimization, finding that stable rank is a reasonable approximation when the user (or item) embeddings are similar, i.e. strong alignment. We then establish a warm-start training strategy which uses stable rank regularization during early phases of training, and other negative sampling-based regularization during the end of training.


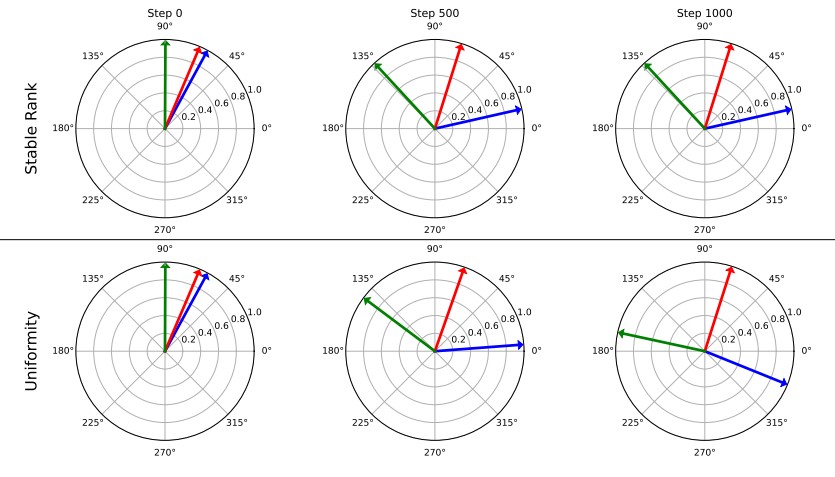

Figure 2: Example of Vectors Optimized for Stable Rank and Uniformity.

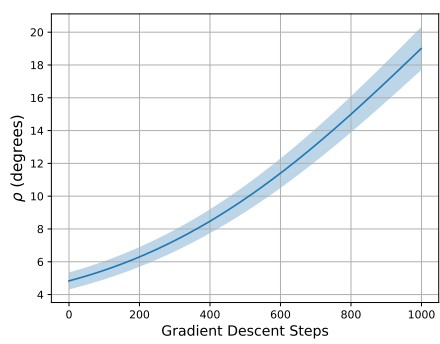

Figure 3: Angle $\rho$ between $\nabla_{\mathbf{u}}L_{\mathbf{srank}}$ and Angle between $\nabla_{\mathbf{u}}L_{\mathbf{uniformity}}$ across gradient descent steps. Fill represents std across three randomly initialized user matrices.

*5.2.1  Relationship Between Stable Rank and Uniformity Optimization.* To establish a connection between stable rank and uniformity, we start with a motivating example where we optimize uniformity and stable rank for three random user vectors in 2-D space. Then, we expand the analysis and look at the angle formed by the uniformity and stable rank gradients, using Equation (12) and (13), for a set of 1000 users in 32-D space. In the higher-dimensional setting, we instantiate the user embeddings such that the angle between pairs of embeddings are within a small angle, $\theta = 1°$, simulating strong alignment. Together, the results highlight that stable rank and uniformity approximate each other well when the vectors are close, and only deviate once the vectors become more uniform.

$$\nabla_u L_{\text{uniformity}} = \frac{-4 \sum_{u'} e^{-2\|\mathbf{u}-\mathbf{u}'\|_2^2} (\mathbf{u} - \mathbf{u}')}{\sum_{u'} e^{-2\|\mathbf{u}-\mathbf{u}'\|_2^2}} \qquad (12)$$

$$\nabla_u L_{\text{srank}} = \frac{\sigma_1^2 2\mathbf{u} - \|\mathbf{U}\|_F^2 (2\sigma_1 \psi_1 \omega_1^\top) u}{\sigma_1^4} \qquad (13)$$

In Figure 2, we plot the three user vectors on the unit circle. In early stages of optimization, the angles between the vectors are highly similar across the two losses. However, as the optimizes continues, the angles diverge between losses, where uniformity continues to separate the vectors apart while the stable rank gradient goes to zero. As stable rank aims to attain orthogonal vectors, it is unable to increase the angle between the vectors beyond Step 500 given the vectors would revert back to linear dependence.

To measure the angle between $\nabla_u L_{\text{uniformity}}$ and $\nabla_u L_{\text{srank}}$, we generate our larger user matrix and compute Equations (12) and (13) for all users. Then, the angle between the gradients is,

$$\rho = \arccos\left(\frac{\nabla_u L_{\text{uniformity}} \cdot -\nabla_u L_{\text{srank}}}{\|\nabla_u L_{\text{uniformity}}\| \|\nabla_u L_{\text{srank}}\|}\right) \qquad (14)$$

across the gradient descent steps. A negative is applied to $\nabla_u L_{\text{srank}}$ as the objective is maximized. In Figure 3, the uniformity and stable rank gradients are highly similar for a majority of the optimization process, and only begin to deviate in the later stages of the optimization. This demonstrates that the smaller example in Figure 3

translates to larger user matrices. We use this insight to inform how we utilize stable rank during training, as described in our section.

*5.2.2  Warm-Start Training Strategy.* As stable rank regularization cannot directly replace uniformity and fully approximate negative sampling, we focus on identifying periods of training where stable rank can be most impactful. Based on the DirectAU trajectories observed in Figure 1, training typically involves three phases focused on alignment, stable rank, and then uniformity. Early in training, alignment dominates the DirectAU loss, leading to an initial collapse in stable rank. Once alignment is achieved, training promotes stable rank, ensuring interacted pairs are closer than non-interacted pairs. Near the end of training, the stable rank is high, however the uniformity continues to push non-interacted pairs apart. Notably, BPR generally only possesses an alignment phase given the weak regularization induced by one negative sample.

To utilize these insights, we propose a warm-start strategy that uses stable rank regularization, in place of negative sampling or uniformity, during the alignment and stable rank phases. Stable rank regularization is then replaced with negative sampling or uniformity regularization during the uniformity phase. To determine the transition point, we employ an early stopping strategy where a decrease in validation performance signals to switch. This is motivated by the fact that once stable rank becomes ineffective, the alignment term will dominate the loss and reduce performance. This early stopping approach accounts for noise by incorporating patience, ensuring the stable rank phase is complete. Given the stable rank computation is significantly faster than more expensive losses, like uniformity, we expect speed-ups in overall training to be roughly proportion with the relative number of epochs warmed up via stable rank. Additionally, since stable rank serves as an inexpensive proxy for negative sampling, we expect lightweight losses which focus on alignment, e.g. BPR, to benefit in performance from stable rank regularization with relatively small runtime costs.

## 6  Empirical Analysis

In this section, we perform a series of empirical analyses to validate the benefit of stable rank regularization and determine how we can

**Table 2: Comparison between MF models trained with DirectAU and DirectAU + Stable Rank warm-start. Test Recall@20 and NDCG@20 are reported, with the runtime measured over training process. We provide the difference between standard and warm start training processes (Stable Rank - Standard) and the percent differences (change from Standard). Stable Rank shown as effective given all datasets receive significant training speedups with no performance loss.**

| DirectAU | Recall@20 | NDCG@20 | Runtime (min) | Recall@20 | NDCG@20 | Runtime (min) |
|---|---|---|---|---|---|---|
| | | MovieLens1M | | | Gowalla | |
| **Standard** | $24.9 \pm 0.0$ | $23.3 \pm 0.1$ | $58.3 \pm 0.7$ | $18.4 \pm 0.1$ | $13.3 \pm 0.1$ | $244.4 \pm 1.9$ |
| **+ Stable Rank** | $25.0 \pm 0.1$ | $23.6 \pm 0.0$ | $40.1 \pm 5.1$ | $18.3 \pm 0.1$ | $13.2 \pm 0.1$ | $148.8 \pm 10.7$ |
| **Difference (% Diff.)** | ↑ 0.1 (0.40%) | ↑ 0.3 (1.29%) | ↓ 18.2 (-31.22%) | ↓ 0.1 (-0.54%) | ↓ 0.1 (-0.75%) | ↓ 95.6 (-39.12%) |
| | | Yelp2018 | | | AmazonBook | |
| **Standard** | $10.6 \pm 0.1$ | $7.1 \pm 0.0$ | $310.9 \pm 3.2$ | $10.5 \pm 0.1$ | $7.8 \pm 0.1$ | $426.6 \pm 9.3$ |
| **+ Stable Rank** | $10.7 \pm 0.1$ | $7.1 \pm 0.0$ | $174.6 \pm 20.2$ | $10.6 \pm 0.1$ | $7.8 \pm 0.1$ | $145.0 \pm 15.7$ |
| **Difference (% Diff.)** | ↑ 0.1 (0.94%) | 0.0 (0.00%) | ↓ 136.2 (-43.81%) | ↑ 0.1 (0.95%) | 0.0 (0.00%) | ↓ 281.6 (-66.03%) |

improve CF training. This leads to three core research questions: (**RQ1**) How effective is our stable rank warm-start strategy at expediting training in computationally expensive settings?, (**RQ2**) How well does stable rank approximate negative sampling, and can it be used to bolster lightweight losses?, and (**RQ3**) What datasets benefit most from stable rank warm-start strategy?

## 6.1 Experimental Setup

*6.1.1 Datasets.* Our experiments are performed on four common benchmarks including MovieLens1M [14], Gowalla [5], Yelp2018 [41], AmazonBook [27]. The dataset statistics are provided in Table 5 of the Appendix.

*6.1.2 Models and Loss Functions.* We focus on applying the stable rank warm-up strategy to MF, trained with BPR, SSM, and DirectAU. We additionally include experiments training LightGCN with DirectAU given together this combination has been shown to produce state of the art results on many tasks. We provide details on hyperparameters and tuning in Appendix C.

*6.1.3 Evaluation Methods/Metrics.* To assess the quality of the ID embedding tables learned under the different model and loss combinations, we look at both performance and training time. For performance, we utilize Recall@20 and NDCG@20, while for runtime we report the time to perform forward and backward passes, given these are the factors which vary between architectures. Exact details on evaluation and implementation are provided in Appendix C.

**Table 3: Comparison between LightGCN models trained with DirectAU and DirectAU + Stable Rank warm-start. Percent differences are shown for NDCG@20 performance and runtime. LightGCN benefits similar to MF with significant improvements in runtime while retaining high performance.**

| Dataset | NDCG@20 % Diff. | Runtime % Diff. |
|---|---|---|
| MovieLens1M | $-0.47 \pm 0.66\%$ | $-35.02 \pm 25.87\%$ |
| Gowalla | $-0.51 \pm 0.36\%$ | $-37.75 \pm 7.19\%$ |
| Yelp2018 | $0.51 \pm 0.71\%$ | $-42.61 \pm 12.99\%$ |
| AmazonBook | $4.69 \pm 2.64\%$ | $-10.27 \pm 16.88\%$ |

## 6.2 Results

**(RQ1) Expediting Training with Stable Rank.** In Table 2, we compare the performance and runtime of MF models trained with vanilla DirectAU and those employing DirectAU with a stable rank warm-start. Across all datasets, the stable rank warm-start significantly reduces runtime while maintaining comparable Recall@20 and NDCG@20 metrics. On average, our approach allows MF models to train 45.93% faster, with speed-ups reaching up to 65.9% on AmazonBook. This improvement stems from two core properties: (i) the stable rank calculation is significantly cheaper than uniformity, accelerating epochs in the alignment or stable rank phases, and (ii) stable rank regularization mitigates the initial stable rank collapse,

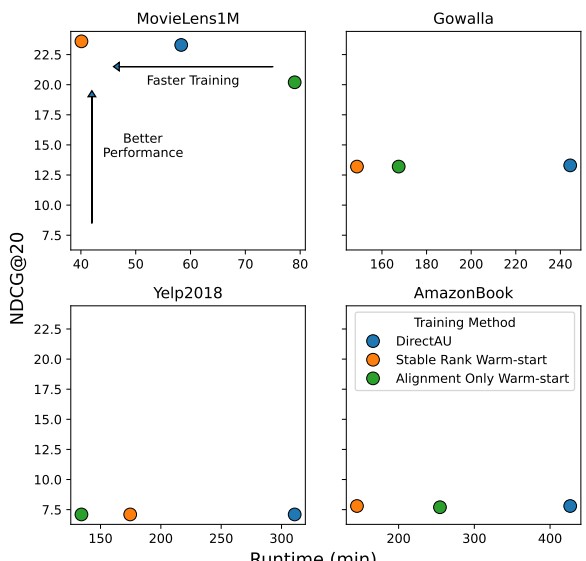

**Figure 4: Ablating the Stable Rank warm-start with Alignment Only warm-start. Each dot represents the NDCG@20 and runtime for the respective training strategy. While all methods attain similar performance, both DirectAU and Alignment Only Warm-start significantly increase runtime.**

**Table 4: Comparison between MF models trained with BPR, and BPR + Stable Rank warm-start. Stable rank is shown to significantly increase performance of BPR with minimal increases in runtime relative to more expensive loss functions.**

| BPR | Recall@20 | NDCG@20 | Runtime (min) | Recall@20 | NDCG@20 | Runtime (min) |
|---|---|---|---|---|---|---|
| | | MovieLens1M | | | Gowalla | |
| Standard | 22.0 ± 0.1 | 19.4 ± 0.0 | 1.2 ± 0.2 | 12.7 ± 0.0 | 9.2 ± 0.1 | 1.8 ± 0.2 |
| + Stable Rank | 22.3 ± 0.0 | 19.8 ± 0.1 | 2.0 ± 0.5 | 13.6 ± 0.1 | 9.9 ± 0.1 | 3.2 ± 1.2 |
| Difference (% Diff) | ↑ 0.3 (1.36%) | ↑ 0.4 (2.06%) | ↑ 0.8 (66.67%) | ↑ 0.9 (7.09%) | ↑ 0.7 (7.61%) | ↑ 1.4 (77.78%) |
| | | Yelp2018 | | | AmazonBook | |
| Standard | 7.1 ± 0.1 | 4.7 ± 0.1 | 2.2 ± 0.1 | 6.6 ± 0.1 | 4.6 ± 0.0 | 18.0 ± 0.7 |
| + Stable Rank | 8.3 ± 0.0 | 5.5 ± 0.0 | 5.4 ± 1.3 | 8.0 ± 0.1 | 5.6 ± 0.1 | 30.7 ± 1.7 |
| Difference (% Diff) | ↑ 1.2 (16.90%) | ↑ 0.8 (17.02%) | ↑ 3.2 (145.45%) | ↑ 1.4 (21.21%) | ↑ 1.0 (21.74%) | ↑ 12.7 (70.56%) |

reducing the number of epochs needed for DirectAU to achieve sufficient uniformity. An ablation study shown in Figure 4 supports these findings where we compare an *alignment-only* warm-start strategy –disabling all regularization in the initial phases – with our stable rank approach. Across nearly all datasets, the alignment warm-start, despite retaining similar performance, takes significantly longer than stable rank to converge.

In Table 3, we report performance and runtime metrics for Light-GCN, revealing similar improvements without performance loss, averaging a 31.4% speed-up across datasets. This result suggests that rank collapse observed in message passing for contrastive learning [38] is also relevant to CF. Notably, the standard deviations for LightGCN are higher than for MF due to its faster convergence – LightGCN typically converges within ∼25 epochs for MovieLens1M, compared to ∼100 epochs for MF. As a single LightGCN epoch has a longer runtime than a single MF epoch, fluctuations in the number of warm-start epochs can produce large swings in percent difference. Nonetheless, the runtime reductions consistently point to significant improvements, underscoring the efficacy of our stable rank warm-start strategy in expediting training.

**(RQ2) Approximating Negative Sampling with Stable Rank.**
In Table 4 we compare MF models trained with vanilla BPR and BPR with stable rank warm-start. Across all datasets, BPR with stable rank warm-start exhibits a notable performance boost, with NDCG@20 increasing by an average of 12.1%, and up to 21.2% on AmazonBook. Importantly, these gains come with only a small increase in runtime, usually just a few extra minutes of computation. To understand the relationship between these performance improvements and negative sampling, we also conducted experiments on SSM, presented in Table 6 of the Appendix. While stable rank regularization offers less benefit to SSM, since SSM is already a cost-effective approximation for full negative sampling, BPR with stable rank regularly achieves similar or better performance. For instance, on Yelp and Gowalla datasets, BPR with stable rank improves over SSM by 22.2% and 3.1% in NDCG@20, respectively. Additionally, on MovieLens1M and Gowalla, BPR with stable rank decreaes runtimes by 60.78% and 48.39% as compared to SSM, demonstrating that our warm-start strategy encodes useful negative sampling signal.

**(RQ3) Stable Rank Warm-start Effectiveness with Different Dataset Properties** While our previous analysis focused on different losses for a fixed dataset, we now examine trends between

datasets for a given loss. As shown in Table 4 and Table 2, our proposed method behaves differently across datasets. Specifically, for MF models trained with DirectAU and BPR, the most substantial speed-ups and performance gains occur with AmazonBook, as mentioned in RQ1 and RQ2. According to Table 5, AmazonBook is the sparsest dataset with a large item set. Conversely, Movie-Lens1M shows the least benefit and is extremely dense with a small item set, only displaying speed-ups of 31.2% and performance improvements of 2.1%. From a negative sampling perspective, sparser datasets with larger item sets require significantly more negative samples to achieve similar levels of regularization as compared to denser datasets with smaller item sets. This is evident in Figure 1, where MovieLens1M trained with BPR shows recovery from the initial stable rank collapse with just one negative sample, while AmazonBook's trajectory strictly decreases. Thus, stable rank regularization tends to be more beneficial for large, sparse datasets, which are commonly seen in real-world systems. For LightGCN, the trend differs due to the runtime bottleneck in the message passing component, which scales with the edge set. Consequently, reducing the number of epochs for LightGCN on denser datasets yields larger runtime improvements, as seen on MovieLens1M and Yelp.

## 7 Conclusion

In this work, we addressed scalability challenges within CF methods by investigating the properties and training trajectories of ID embedding tables under various learning strategies. Through both empirical and theoretical analyses, we revealed an intrinsic link between the singular values of these tables and different CF loss functions. Moreover, leveraging the relationship between BPR, SSM, and DirectAU, rooted in the level of regularization induced by negative sampling, we proposed a an efficient stable rank regularization which promotes similar training signals as full negative sampling. To operationalize our proposed method, we also proposed a warm-start strategy which optimizes for stable rank during the early training phases, significantly improve embedding quality and efficiency. These findings both enhance our fundamental understanding of CF-based recommender systems, while also broadening their applicability to large-scale environments by mitigating computational overhead. Furthermore, as the method is model and loss function agnostic, it can be easily combined with more modern CF learning paradigms, as well as more mature production pipelines.

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

# A Theoretical Analysis for Alignment and Uniformity

The goal of this analysis is to understand how alignment (no negative sampling) and uniformity (full negative sampling) induce different stable rank properties in the user and item embedding matrices. We begin by studying the gradient descent process for the user embedding matrix as trained by the alignment loss function, represented as the euclidean distance between user-item pairs. Then, we will consider the uniformity loss term, and perform a similar style of analysis.

## A.1 Proof for Theorem 1 - Alignment Induces Rank Collapse

We begin by understanding how the alignment objective modifies the singular values of the user (or item) matrices. Without loss of generality, we focus on the singular values of the user matrix, however the same logic can be applied to the item matrix by flipping the user and item notation.

Assume there is a user matrix $\mathbf{U} \in \mathbb{R}^{n \times d}$ and an item matrix $\mathbf{I} \in \mathbb{R}^{m \times d}$. Additionally, assume there are a set of historic interactions $\mathcal{E}$, where $(j, l) \in \mathcal{E}$ denotes that a user $j$ interacts with item $l$. Then, the alignment objective over the user and item pairs is:

$$L_{\text{align}} = \sum_{(j,l) \in E} \|\mathbf{u}_j - \mathbf{i}_l\|_2^2.$$

As we can group terms within $L_{\text{align}}$ according to particular items, similar to batching within gradient-based optimization, we focus on a single arbitrary item $\mathbf{i}$ with $r$ interacted users. Through this notation, a user $\mathbf{u}_j \in \{\mathbf{u}_1, ..., \mathbf{u}_r\}$ is attached to item $\mathbf{i}$. The alignment loss for this subset is expressed as:

$$L_{\text{align}} = \sum_{j=1}^{r} \|\mathbf{u}_j\|_2^2 - 2\mathbf{u}_j^\top \mathbf{i} + \|\mathbf{i}\|_2^2.$$

The partial derivative for the alignment loss with respect to a user $\mathbf{u}_j$ is give by,

$$\frac{\partial \mathcal{L}}{\partial \mathbf{u}_j} = 2(\mathbf{u}_j - \mathbf{i})$$

leading to a recursive gradient descent update rule of,

$$\mathbf{u}_j^{(t+1)} = \mathbf{u}_j^{(t)} - 2\eta(\mathbf{u}_j^{(t)} - \mathbf{i}).$$

where $\eta$ is the learning rate and $\mathbf{u}_j^{(t+1)}$ is the $j$-th user's embedding vector at time $t$. To attain a closed-form solution of the embedding for $\mathbf{u}_j$, the recurrence relation is expanded, leading to:

$$\mathbf{u}_j^{(t)} = (1 - 2\eta)^t \mathbf{u}_j^{(0)} + 2\eta \sum_{k=0}^{t-1} (1 - 2\eta)^k \mathbf{i}.$$

This result can be further simplified by summing the geoemtric series, leading to,

$$\mathbf{u}_j^{(t)} = (1 - 2\eta)^t \mathbf{u}_j^{(0)} + \left(1 - (1 - 2\eta)^t\right) \mathbf{i}.$$

The update equation for each user can be consolidated into matrix-form by recognizing the equation as a weighted sum of the original user embeddings, and the item embeddings. Then, the update rule becomes,

$$\mathbf{U}^{(t)} = (1 - 2\eta)^t \mathbf{U}^{(0)} + (1 - (1 - 2\eta)^t)(\mathbf{1} \otimes \mathbf{i}^\top)$$

where $\mathbf{U}^{(t)}$ denotes the user matrix at gradient descent step $t$, $\mathbf{1}$ represents a ones vector of size $r$ and $\otimes$ is the outer product.

The singular values of the matrix $\mathbf{U}^{(t)}$ can be directly inferred based on the update rule. First, we assume that $\mathbf{U}^{(0)}$, the original user embeddings, has a singular value decomposition (SVD), where $\mathbf{U}^{(0)} = \sum_{j=1}^{min(r,d)} \sigma_j^{(0)} \alpha_j^{(0)} \beta_j^{(0)}$, and all $\sigma_j^{(0)} > 0$. That is, $\mathbf{U}^{(0)}$ is full rank. Then, we can study the singular values of the two terms in the update rule. The $j$-th singular values for the two terms of $\mathbf{U}^{(t)}$ are given by:

$$\sigma_j((1 - 2\eta)^t \mathbf{U}^{(0)}) = (1 - 2\eta)^t \sigma_j^{(0)}$$

$$\sigma_j((1 - (1 - 2\eta)^t)(\mathbf{1} \otimes \mathbf{i}^\top)) = \begin{cases} (1 - (1 - 2\eta)^t)\sqrt{r}\|\mathbf{i}\|_2, & j = 1 \\ 0, & j \neq 1 \end{cases}.$$

Using Weyl's inequality, we can then bound the $j$-th singular value of $U^{(t)}$ as:

$$\sigma_j(\mathbf{U}^{(t)}) \leq \sigma_j((1 - (1 - 2\eta)^t)(\mathbf{1} \otimes \mathbf{i}^\top)) + \sigma_1((1 - 2\eta)^t \mathbf{U}^{(0)})$$
$$= \begin{cases} (1 - (1 - 2\eta)^t)\sqrt{r}\|\mathbf{i}\|_2, & \text{if } j = 1 \\ 0, & \text{if } j \neq 1 \end{cases} + (1 - 2\eta)^t \sigma_1^{(0)}$$
$$(15)$$

leveraging the relationship between the 2-norm of a matrix and the dominant singular value. Additionally, $\sigma_1^{(0)}$ is the largest singular value of the original sampled user matrix. To understand how the singular values are changing, we can compare $\frac{\sigma_1^{(0)}}{\sigma_2^{(0)}}$ with $\frac{\sigma_1^{(t)}}{\sigma_2^{(t)}}$, denoted $\Delta^{(t)}$, by dividing the two quantities to attain a relative scaling. While Weyl's inequality provides an upper bound, the exact values are dictated by exact properties of the user and item matrices. As these cannot be directly expressed, we instead introduce $\lambda_1 \leq 1$ and $0 < \lambda_2 \leq 1$ as scalars on the first and second singular values to account for instance where they are not at the upper bound. Then,

$$\Delta^{(t)} = \frac{\sigma_1^{(0)} \lambda_2 (1 - 2\eta)^t}{\lambda_1 (1 - (1 - 2\eta)^t)\sqrt{r}\|\mathbf{i}\|_2 + (1 - 2\eta)^t \sigma_1^{(0)}}$$

Assuming that $0 < \eta < 0.5$, otherwise the representations either do not change, or immediately collapse to $\mathbf{i}$, the term $(1 - (1 - 2\eta)^t)\sqrt{r}\|\mathbf{i}\|$ is strictly positive. Letting $\lambda_1 \approx \lambda_2$, we cancel these terms and conclude that $\Delta^{(t)} < 1$, and the gap between the first and second largest singular value exponentially increases as a function of $t$. Thus, one can expect that after sufficient iterations, alignment will induce rank collapse on the subset of the user matrix. □

## A.2 Proof for Theorem 2 - Uniformity Promotes Higher Rank

Using our previously established result and analysis techniques, we are now going to study uniformity. The goal will be to be able to express a similar measure for the gap between the two largest singular values. We will assume a batch of $r$ users with $d$ features,

and study how user matrix $\mathbf{U}$ is updated. Similar logic can be applied to the item matrix, $I$. We begin by specifying the uniformity loss as

$$L_{\text{uniformity}} = \log\left(\sum_j^r \sum_{j' \neq j}^r e^{-2\|\mathbf{u}_j - \mathbf{u}_{j'}\|_2^2}\right).$$

To perform a similar analysis based on matrix computation, let us first represent the uniformity term through matrix computation. We will begin through a similar expansion used in the alignment computation where:

$$\|\mathbf{u}_j - \mathbf{u}_{j'}\|_2^2 = \|\mathbf{u}_j\|_2^2 - 2\mathbf{u}_j^\top \mathbf{u}_{j'} + \|\mathbf{u}_{j'}\|_2^2.$$

Given the log transform on the uniformity term is monotonic and simply changes the scale of the gradient update, we remove it for simplicity. Then, the uniformity loss is:

$$L_{\text{uniformity}} = \sum_j^r \sum_{j' \neq j}^r e^{-2(\|\mathbf{u}_j\|_2^2 - 2\mathbf{u}_j^\top \mathbf{u}_{j'} + \|\mathbf{u}_{j'}\|_2^2)}.$$

We can the compute the gradient with respect to a user $u_j$ as:

$$\nabla_{u_j} L_{\text{uniformity}} = \sum_{j' \neq j}^r -4e^{-2(\|\mathbf{u}_j\|_2^2 - 2\mathbf{u}_j^\top \mathbf{u}_{j'} + \|\mathbf{u}_{j'}\|_2^2)}(\mathbf{u}_j - \mathbf{u}_{j'}). \quad (16)$$

We can then use the fact that the rows of $\mathbf{U}$ are normalized, leading to

$$\nabla_{u_j} L_{\text{uniformity}} = \sum_{j' \neq j}^r -4e^{-2(2 - 2\mathbf{u}_j^\top \mathbf{u}_{j'})}(\mathbf{u}_j - \mathbf{u}_{j'}).$$
$$= -4e^{-4} \sum_{j' \neq j}^r e^{4\mathbf{u}_j^\top \mathbf{u}_{j'}}(\mathbf{u}_j - \mathbf{u}_{j'}) \quad (17)$$

The gradient descent update for a user $\mathbf{u}_j$ is:

$$\mathbf{u}_j^{(t+1)} = \mathbf{u}_j^{(t)} + \eta 4e^{-4} \sum_{j'}^r e^{4\mathbf{u}_j^\top \mathbf{u}_{j'}}(\mathbf{u}_j - \mathbf{u}_{j'})$$

Now, let $\mathbf{U}\mathbf{U}^\top$ be the Gram matrix of $\mathbf{U}$, $\mathbf{G}$. Likewise, let $\delta\mathbf{U}_k \in \mathbb{R}^{n \times n}$ be the matrix of pairwise differences for the $k$-th element in the user embeddings. Then, the update for the full matrix $\mathbf{U}$ is

$$\mathbf{U}^{(t+1)} = \mathbf{U}^{(t)} + \eta 4e^{-4}[(e^{4G} \otimes \delta\mathbf{U}_0)\mathbf{1}^{n \times 1}, ..., (e^{4G} \otimes \delta\mathbf{U}_f)\mathbf{1}^{n \times 1}] \quad (18)$$

The second term in the equation denotes the element-wise product of the similarities from the exponential gram matrix and the pairwise differences. $\mathbf{1}^{n \times 1}$ is used to sum across all pairwise elements for a given user, weighted by the similarity. Finally, all of these elements are stacked for each dimension of the embedding.

The singular values can be expressed using Weyl's inequality as

$$\sigma_j(\mathbf{U}^{(t+1)}) \leq \sigma_j(\eta 4e^{-4}[(e^{4G} \otimes \delta\mathbf{U}_0)\mathbf{1}, ..., (e^{4G} \otimes \delta\mathbf{U}_f)\mathbf{1}]) + \sigma_1(\mathbf{U}^{(t)}) \quad (19)$$

In order to attain a closed form solution, we make the assumption that each of the dimensions from $0$ to $f$ have the same pairwise distance matrix $\delta\mathbf{U}$. Then, the expression can be simplified as,

$$\sigma_j(\mathbf{U}^{(t+1)}) \leq \sigma_j(\eta 4e^{-4}(e^{4G} \otimes \delta\mathbf{U}_0)\mathbf{1}^{n \times d}) + \sigma_1(\mathbf{U}^{(t)})$$
$$= \eta 4e^{-4}\sigma_j((e^{4G} \otimes \delta\mathbf{U})\mathbf{1}^{n \times d}) + \sigma_1^{(t)}$$
$$= \eta 4e^{-4}\sigma_j(e^{4G} \otimes \delta\mathbf{U})\sigma_1(\mathbf{1}^{n \times d}) + \sigma_1^{(t)} \quad (20)$$
$$= \eta 4e^{-4}\sqrt{rd}\sigma_j(e^{4G} \otimes \delta\mathbf{U}) + \sigma_1^{(t)}$$

We thus need to analyze $\sigma_j(e^{4G} \otimes \delta\mathbf{U})$. We will begin by approximating $e^{4G}$ as a first order Taylor series of $e^X = \mathbf{I} + \mathbf{X}$ to linearize the relationship. Then $\sigma_j(e^{4G} \otimes \delta\mathbf{U}) \approx \sigma_j((\mathbf{I} + 4\mathbf{G}) \otimes \delta\mathbf{U})$. Using the properties of singular values of hadamard products, we can lower bound the relationship as $\sigma_i(\mathbf{A} \otimes \mathbf{B}) \geq \sigma_n(\mathbf{A})\sigma_i(\mathbf{B})$. Then, the lower bound is:

$$\sigma_i((\mathbf{I} + 4\mathbf{G}) \otimes \delta\mathbf{U}) \geq \sigma_n((\mathbf{I} + 4\mathbf{G}))\sigma_i(\delta\mathbf{U})$$
$$= (1 + 4\sigma_r(\mathbf{G}))\sigma_i(\delta\mathbf{U}) \quad (21)$$
$$> \sigma_i(\delta\mathbf{U})$$

assuming that $G$ is full rank with smallest singular value greater than 0. Applying the lower bound:

$$\sigma_j(\mathbf{U}^{(t+1)}) \leq \eta 4e^{-4}\sqrt{rd}\sigma_j(\delta\mathbf{U}) + \sigma_1^{(t)} \quad (22)$$

We can then compute the ratio between $\frac{\sigma_1(\mathbf{U}^{(t)})}{\sigma_2(\mathbf{U}^{(t)})}$ and $\frac{\sigma_1(\mathbf{U}^{(t+1)})}{\sigma_2(\mathbf{U}^{(t+1)})}$, $\Delta^{(t)}$ to measure how the singular values change with uniformity optimization, again introducing $\lambda_1, \lambda_2$ as we did in the proof on alignment to account for the divergence from the upper bound. Then,

$$\Delta^{(t)} = \frac{\sigma_1(\mathbf{U}^{(t)})\lambda_2(\alpha\sigma_2(\delta\mathbf{U}^t) + \sigma_1^{(t)})}{\sigma_2(\mathbf{U}^{(t)})\lambda_1(\alpha\sigma_1(\delta\mathbf{U}^t) + \sigma_1^{(t)})} \quad (23)$$

where $\alpha = \eta 4e^{-4}\sqrt{rd}$. We can then solve for when $\Delta^{(t)} > 1$ indicating that the first and second singular values have a smaller relative gap. This occurs when $\frac{\sigma_1(\mathbf{U}^{(t)})}{\sigma_2(\mathbf{U}^{(t)})} > \frac{\sigma_1(\delta\mathbf{U}^{(t)})}{\sigma_2(\delta\mathbf{U}^{(t)})}$, assuming $\lambda_1 \approx \lambda_2$.

If we study rank-2 matrices, considering a particular rank-2 matrix $\mathbf{M}$, $\frac{\sigma_1(M^t)}{\sigma_2(M^t)}$ is the condition number $\kappa(\mathbf{M}^t)$, and we want to assess when $\kappa(\mathbf{M}^t) > \kappa(\delta\mathbf{M}^t)$ where $\delta\mathbf{M}$ is the pairwise difference matrix where all element-wise distances are a constant. We will also assume that the vectors are close in proximity, modeled as

$$\mathbf{M} = \begin{pmatrix} \cos(\theta) & \sin(\theta) \\ \cos(\theta + \epsilon) & \sin(\theta + \epsilon) \\ \vdots & \vdots \\ \cos(\theta + (r-1)\epsilon) & \sin(\theta + (r-1)\epsilon) \end{pmatrix} \quad (24)$$

Then, we have that $\sigma_1(\mathbf{M}) = \sqrt{r}$ from the Frobenius norm of $M$, while $\sigma_2(\mathbf{M})$ scales as $\epsilon\sqrt{r}$ given the second singular value is proportional to the spread in angle between vectors. Thus, $\kappa(\mathbf{M}^t) \approx \frac{1}{\epsilon}$, indicating an increased gap in the two singular values as $\epsilon$ becomes smaller, approaching a collapse in the second singular value. Similarly, $\sigma_1(\delta\mathbf{M})$ will scale with the maximum distance between the furthest vectors in $U$, which is equivalent to $2|\sin((r-1)\epsilon/2)|$, i.e. the angle spanned by the vectors with angles $\theta$ and $\theta + (r-1)\epsilon$, when $\epsilon$ is small. Similarly, we can expect $\sigma_2(\delta\mathbf{M})$ to scale as $\epsilon\sqrt{r}$ given

the dependence on the spread of angles and number of vectors. With $\epsilon$ small, the small angle approximation of $2|sin((r-1)\epsilon/2)|$ is $(r-1)\epsilon$, and $\kappa(\delta\mathbf{M}^t) \approx \frac{(r-1)\epsilon}{\epsilon\sqrt{r}} = \frac{(r-1)}{\sqrt{r}}$. For $\lim_{\epsilon\to 0}$ with a fixed $r$, it is clear that $\kappa(\mathbf{M}^t) > \kappa(\delta\mathbf{M}^t)$, indicating a decrease in the gap in the first and second singular values with uniformity optimization, increasing stable rank.

□

## B Dataset Statistics

For the datasets used throughout the paper, we provide their statistics in Table 5. We provide the number of users and items, as well as the number of interactions. Additionally, we compute the density of each dataset as # Interactions/(# Users × # Items).

### Table 5: Dataset Statistics.

| Dataset | # Users | # Items | # Interactions | Density |
|---------|---------|---------|----------------|---------|
| MovieLens1M | 6,040 | 3,629 | 836,478 | 3.82% |
| Gowalla | 29,858 | 40,981 | 1,027,370 | 0.08% |
| Yelp2018 | 31,668 | 38,048 | 1,561,406 | 0.13% |
| AmazonBook | 52,643 | 91,599 | 2,984,108 | 0.06% |

## C Additional Setup Details on Experiments

In this section, we provide additional experiment setup details for the analyses performed within the paper. Note that the details apply to both the motivating experiment given on stable rank trajectories, as well as the warm-start experiments. We start by giving additional details on the hyper-parameter tuning process, then provide information on the evaluation process and implementation details.

### C.1 Hyperparameters and Tuning

We primarily focus on MF throughout the paper, with LightGCN added in the warm-start experiments. The embedding tables in both cases are initialized using PyTorch's standard unit normal distribution. We use cross validation to choose the best model, searching over learning rates $\{0.1, 0.01, 0.001\}$ and weight decays $\{1e^{-4}, 1e^{-6}, 1e^{-8}\}$. The embedding dimensions are kept at 64. For LightGCN, we set a depth of 3. The models are trained for up to 400 epochs, with early stopping employed over validation NDCG@20. A patience value of 20 epochs is used, based on our sensitivity study performed in Appendix D.3. For experiments which utilize SSM, we set a negative sampling ratio of 20. For DirectAU, we additionally cross-validate $\gamma$ values from $\{1.0, 2.0, 5.0\}$, as recommender in the original paper [35]. The batch size for BPR, SSM, and DirectAU training are set to roughly maximize size that can fit within memory, which is 16384 for BPR and SSM, and 4096 for DirectAU. When using the stable rank regularization, we hyper-parameter tune $\gamma_{sr}$ from $\{0.05, 0.1, 0.2\}$, however we find that 0.1 tends to work well for most datasets. The train/val/test splits are random and use 80%/10%/10% of the data.

### C.2 Evaluation

To evaluate our models, we look at recall@K, NDCG@K, and runtime in minutes. We use standard definitions for recall@K and

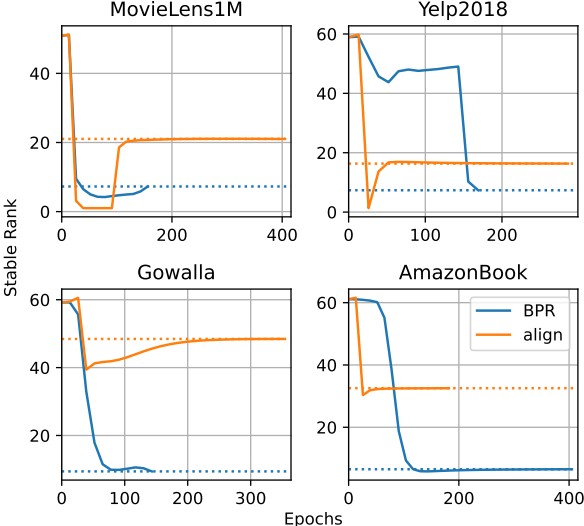

**Figure 5: Stable Rank Trajectories of Item Matrices. The blue line denotes BPR, and the orange line denotes DirectAU. The dotted line denotes the final stable rank of the item matrices after training. DirectAU produces higher rank user matrices.**

NDCG@K, except that we let $k = \min(k, N(u))$, where $N(u)$ is the number of elements a user $u$ interacts with. This way, each user's recall and NDCG value can span the full range from 0 to 1. For runtime, we specifically focus on timing the forward and backward passes of the different methods. Thus, we do not include evaluation given it is constant between methods. Moreover, evaluation can be approximated with fewer samples, or performed on a subset of epochs, and thus naturally sped up if significant runtime costs are incurred.

### C.3 Implementation

The loss functions, as well as the MF training process, and implemented within vanilla PyTorch. LightGCN is implemented using PyTorch Geometric. Data loading and batching is additionally implemented with PyTorch Geometric's dataloader. We use approximate negative sampling for BPR and SSM, as seen in PyG's documentation for their LinkNeighborLoader, meaning there is a small chance some negative samples may be false negatives. The models are trained on single Tesla P100s with 32GB of RAM, via Google Cloud.

## D Additional Experimental Results

In this section, we provide supplemental results to the experiments performed in the main text.

### D.1 Item Stable Rank Trajectories

We include the stable rank trajectories for the item set in Figure 5, denoting similar trends to those seen in the user embedding table for both BPR and DirectAU.

**Table 6: Comparison between SSM models trained with standard training, as well as with Stable Rank regularization as a warm-start process. Reported are Recall@20, NDCG@20 on the held-out test set, with the runtime measured over the training process. Stable rank shown as less effective for SSM given the loss function already utilizes a larger number of negative samples (20 negative samples).**

| Loss | Recall@20 | NDCG@20 | Runtime (min) | Recall@20 | NDCG@20 | Runtime (min) |
|---|---|---|---|---|---|---|
| | | **MovieLens1M** | | | **Gowalla** | |
| **Standard** | 23.4 ± 0.1 | 20.8 ± 0.2 | 5.1 ± 0.8 | 13.0 ± 0.3 | 9.6 ± 0.2 | 6.2 ± 0.0 |
| **+ Stable Rank** | 23.3 ± 0.0 | 20.9 ± 0.1 | 5.1 ± 0.5 | 13.1 ± 0.4 | 9.8 ± 0.3 | 6.3 ± 0.0 |
| **Difference** | ↓ 0.1 (-0.43%) | ↑ 0.1 (0.48%) | 0.0 (0.00%) | ↑ 0.1 (0.77%) | ↑ 0.2 (2.08%) | ↑ 0.1 (1.61%) |
| | | **Yelp2018** | | | **AmazonBook** | |
| **Standard** | 6.7 ± 0.7 | 4.5 ± 0.3 | 3.8 ± 0.6 | 8.1 ± 0.5 | 6.1 ± 0.4 | 20.9 ± 2.1 |
| **+ Stable Rank** | 6.8 ± 0.7 | 4.7 ± 0.4 | 6.8 ± 1.3 | 8.1 ± 0.4 | 6.1 ± 0.4 | 22.4 ± 2.2 |
| **Difference** | ↑ 0.1 (1.49%) | ↑ 0.2 (4.44%) | ↑ 3.0 (78.95%) | 0.0 (0.00%) | 0.0 (0.00%) | ↑ 1.5 (7.18%) |

## D.2 Results on SSM

In this section, we provide results when training MF with the SSM loss, using 20 negative samples. Given the relationship between BPR, SSM, and DirectAU, rooted in the number of negative samples acting as weaker or stronger regularization, SSM acts a intermediary between the BPR and DirectAU. The results in Table 6 highlight this fact, where the 20 negative samples already offer a reasonable trade-off of performance versus runtime, and thus do not benefit strongly from stable rank. Discussion on these results are offered in the main text, but at a high-level, BPR with stable rank is able to attain comparable performance to SSM with and without stable rank with significant lower runtime. This result demonstrates the benefit of approximating negative sampling through stable rank, allowing BPR significant performance gains with less computational overhead.

## D.3 Sensitivity to Patience

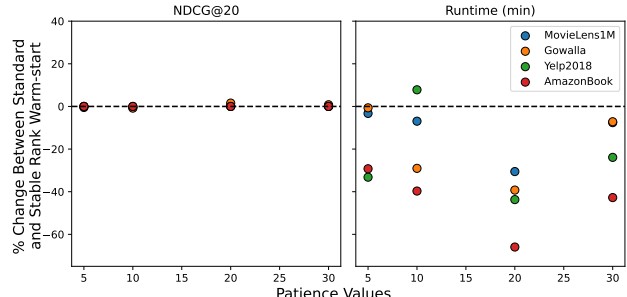

**Figure 6**

We perform a sensitivity analysis on our choice of patience, looking at different patience levels when training MF with DirectAU and DirectAU with stable rank warm-start. In Figure 6, we can see that performance has no significant chances, and stays constant across settings. On the other hand, the runtime speed-ups do have some sensitivity to patience, which tends to be best around 20. Before that, we risk the model early stopping due to noise, and then

requiring more uniformity epochs to attain optimal performance. Moreover, if the patience is set higher to 20, then the optimization risks residing in the stable rank phase of training too long, again requiring more overall uniformity epochs.

