# OpenReview forum: "Understanding and Scaling Collaborative Filtering Optimization from the Perspective of Matrix Rank"
_ACM.org/TheWebConf/2025/Conference — WWW 2025 Oral_

### Official Review · Reviewer_Mvet · 2024-11-22

**Novelty:** 4
**Technical Quality:** 5

**Review:**

This paper presents a novel approach to CF by introducing stable rank regularization as a cost-efficient alternative to traditional negative sampling methods. The authors focus on the relationships between negative sampling, matrix rank, and CF performance, providing both empirical and theoretical insights into how stable rank can enhance the training of RS.

However, I'm not an expert in the theory of Matrix Rank. It seems that the proposed method improves the runtime while retaining high performance.

**Questions:**

1. Why does the rank in Figure 2 have turning points of increase?
2. How do you explain the Alignment Only Warm-start approach converges faster than Stable Rank Warm-start on the Yelp2018 dataset with similar performance?

**Reviewer Confidence:**

1: The reviewer's evaluation is an educated guess

**Scope:**

4: The work is relevant to the Web and to the track, and is of broad interest to the community

---

### Official Review · Reviewer_em8j · 2024-11-22

**Novelty:** 6
**Technical Quality:** 6

**Review:**

This work found that the singular values of the embedding tables are intrinsically linked to CF loss functions through theoretical analysis, and demonstrated the practical benefits of higher stable rank.
Based on this, this paper proposes a warm-start strategy that regularizes the stable rank of the user and item embeddings. The result shows that stable rank regularization during early training phases can promote higher-quality embeddings

**Strength**
1. The evolution of the stable rank of different CF methods during the training process is discussed, and the relationship between the stable rank and the final recommendation effect is exposed.
2. The theoretical analysis in this text is very sound.
3. Experimental results demonstrate the role of stable rank regularization.

**Weakness**

The experiment lacks a stable rank trajectory using stable rank regularization, and it would be desirable to add the NDCG@20 score for whether or not stable rank regularization is used at the same moment. This is very important because this trajectory reflects whether or not stable rank regularization is actually working.

**Questions:**

1. It can be observed that DirectAU ends up exhibiting a raised stable rank in all experimental results. The paper lacks a comparison with the final stable rank using stable rank optimization. Although it is experimentally demonstrated that the use of stable rank exhibits a time advantage, this advantage is only demonstrated for one method, DirectAU. I would like to know if a higher stable rank can be obtained by stable rank regularization.
2. Is the stable rank actually an intuitive showcase of the strengths and weaknesses of the final embedding generated by different CF optimization strategies? Even from that perspective, stable rank is certainly very valuable. However, using this as an optimization objective seems to make it difficult to directly intervene in the effectiveness of the final generated embedding.
3. Analyzed from the point of view of alignment and uniformity, the stable rank exhibits more of what seems to be an optimization for uniformity. This is my personal guess as to why its final performance in terms of recommendations is not very impressive. This is because all these CF loss functions are mainly optimized for uniformity. Optimizing the stable rank does not significantly optimize the uniformity on the basis of these methods.
4. In the Introduction the task considered stable rank optimization as an efficient proxy for negative sampling. But in fact, DirectAU explicitly states in the article that its training is independent of negative samples, so maybe this method is a proxy for uniformity optimization.

**Reviewer Confidence:**

3: The reviewer is confident but not certain that the evaluation is correct

**Scope:**

4: The work is relevant to the Web and to the track, and is of broad interest to the community

---

### Official Review · Reviewer_1L3Y · 2024-12-02

**Novelty:** 5
**Technical Quality:** 5

**Review:**

The paper solves scalability challenges in CF by introducing stable rank regularization with a warm-start strategy. This approach optimizes the stable rank of user and item embeddings during the early training phases. By linking the stable rank of embedding matrices with CF performance, the authors show that stable rank regularization can replace costly negative sampling strategies like BPR, SSM, and DirectAU, offering a computationally efficient alternative.

Pros：

1. The paper proposes the stable rank regularization approach, which offers an innovative perspective and theoretically addresses the scalability issues in large-scale collaborative filtering models, showing significant innovation.

2. The paper combines theoretical analysis and empirical experiments, providing detailed theoretical derivations to demonstrate the relationship between stable rank and model performance. The method's effectiveness is further validated through experiments, enhancing the credibility and practicality of the research.

3. Also, the structure of the paper is clear and logically rigorous. The writing flows smoothly and helps readers better understand the approach.

Cons:
1. The extensive textual explanations may make it difficult for readers to quickly grasp the core concepts, especially for some of the complex theoretical derivations.

2. Although the paper includes some experiments and discussions on SSM, the analysis compared to other methods is still not comprehensive enough.

3. The method has not been sufficiently validated for its applicability to complex deep learning models.

4. The paper contains some spelling mistakes (like in line 208,'consider').

**Questions:**

1. Although the paper explains the method in detail, it lacks enough diagrams to clearly show how stable rank regularization works. It would be helpful if the authors could add more visual aids to make the method easier to understand.

2. The paper mentions SSM and includes some experiments, but the comparison with other methods is not detailed enough. The authors may expand the analysis to provide a deeper comparison.

3. The paper has experiments with LightGCN, but it’s unclear whether the method will work well with other deep learning models. Maybe more tests on complex deep learning models should be done to show if the method can be applied widely.

4. There are spelling errors and minor language issues in the paper. It’s recommended to proofread the paper carefully.

5. The authors should include a section discussing potential limitations and challenges of their approach.

**Reviewer Confidence:**

3: The reviewer is confident but not certain that the evaluation is correct

**Scope:**

4: The work is relevant to the Web and to the track, and is of broad interest to the community